# Factors Influencing Global Variations in COVID-19 Cases and Fatalities; A Review

**DOI:** 10.3390/healthcare8030216

**Published:** 2020-07-17

**Authors:** Osama Abu-Hammad, Ahmad Alnazzawi, Sary S. Borzangy, Abdalla Abu-Hammad, Mostafa Fayad, Selma Saadaledin, Shaden Abu-Hammad, Najla Dar-Odeh

**Affiliations:** 1College of Dentistry, Taibah University, Al Madinah Al Munawara 43353, Saudi Arabia; anazawi@taibahu.edu.sa (A.A.); sbarzanji@taibahu.edu.sa (S.S.B.); dr.mifayad@gmail.com (M.F.); najla_dar_odeh@yahoo.com (N.D.-O.); 2School of Dentistry, University of Jordan, Amman 11942, Jordan; Shadenabuhammad@gmail.com; 3School of Medicine, University of Jordan, Amman 11942, Jordan; abdullah018ju@gmail.com; 4Faculty of Dental Medicine, Al-Azhar University for Boys, Cairo 11751, Egypt; 5Schulich School of Medicine and Dentistry, Western University, London, ON N6A 5C1, Canada; selmasadeldin@gmail.com

**Keywords:** age, COVID-19, genetics, social factors, geographic factors

## Abstract

Since the first cases of the novel corona virus disease (COVID-19) were diagnosed in China, outcomes associated with this infection in terms of total numbers of cases and deaths have varied widely between countries. While some countries had minimal rates of infections and deaths, other countries were hit hard by the pandemic. Countries with highest numbers of cases continued to change over time, but at the time of submission of this article they are: USA, Brazil, Russia, UK, India, Spain, Italy, Peru and Chile. This is in contrary to many countries in the Middle East, Far East, and Africa, which had lower cases or deaths/cases rates. This raised many questions pertaining to this variation. This overview explores the potential factors that contribute to spread, transmission and outcomes of the COVID-19 infection. It also uses an evidence-based approach in reviewing the available most recent literature that tackled the various factors that modify the populations’ response to COVID-19, namely, factors pertaining to population characteristics, environmental and geographic factors.

## 1. Introduction

COVID-19 is a newly emerging disease caused by severe acute respiratory distress syndrome corona virus-2 (SARS-CoV-2), an RNA virus with a sequence of around 29,000 base pairs. Similar to other coronaviruses, such as SARS-CoV and Middle East respiratory distress syndrome corona virus (MERS-CoV), the genetic material is similar to that of bats, implicating the bat as the source of this virus. While the exact source of this infection is not clear yet, first confirmed cases were identified in Wuhan, Hubei, China, in November 2019 [1]. By the early days of July 2020, cases worldwide exceeded 12 million with fatalities in excess of 500,000. While numbers in the United States exceeded 3 million, Brazil had more than 1.7 million cases, India more than 0.7 million cases, Russia more than 0.7 million, and Peru more than 0.3 million cases. China, a heavily populated country, and where the disease was first identified, has been less affected than the above-mentioned countries with approximately 83,000 cases only.

This disease is characterized by the rapid spread across the globe, and it was announced as a Public Health Emergency of International Concern on 30th January 2020. It is now recognized that there are multiple routes of transmission of the virus; direct or indirect contact via respiratory droplets emitted when coughing, sneezing or even talking, and a fecal-oral route of transmission which has also been implicated by many reports [2,3].

The cost of living is undoubtedly the most important driving force to stop the spread of this virus—however, economic consequences are severe. The United Nations’ trade and development agency stated that slowdown in the global economy caused by COVID-19 outbreak is likely to cost more than $1 trillion [4].

As of July 2020, countries that were hit hard by the pandemic in terms of the number of cases include (in descending order): USA, Brazil, India, Russia, Peru, and Chile. Countries with the highest toll of deaths were (in descending order): USA, Brazil, UK, Italy, and Mexico. (Table 1).

While the early statistics of the pandemic showed a high fatality/cases rates of 4–8% in the UK, Spain, and Italy [5], even higher fatality rates were recorded beginning of July 2020 and these occurred in France (17%), the UK (15%), Italy (14%), and Spain (9%).

Worldwide the wide variability in the number of deaths/cases in different countries has attracted attention as to what could cause such variations. As research efforts are intensifying worldwide to clarify ambiguities surrounding this virus and its behavior, including the biological and therapeutic aspects, data are emerging from different official and non-official sources, with a major proportion of this data being unreliable. Identifying risk factors for severe illness and death could allow identification of vulnerable groups most likely to have poor outcomes and choosing the right practices in prevention and treatment [6].

This review employs an evidence-based approach to explore the possible factors that contributed to the high numbers of cases and death rates in specific countries worldwide. It also aims to identify false factors that have wrongly been implicated in accelerating infections and deaths/cases rates by the virus.

Several official websites have provided information on COVID-19, and they constituted the main source for identifying possible factors linked with the spread and transmission of COVID-19. These include: World Health Organization (WHO) (https://covid19.who.int/), Center for Disease Prevention and Control (CDC, Atlanta, GA, USA), European Centre for Disease Prevention and Control (ECDC), Worldometers, Our world in data, Live Science, and Unicef.

## 2. Potential Factors Influencing COVID-19 Case and Death Rates

Several factors were identified as potential factors that may influence infection and death rates. These factors could be grouped into four main categories: Population characteristics, environmental/geographic factors, healthcare policy and virus-related factors. These factors are presented in Figure 1.

### 2.1. Population Characteristics

There are many population characteristics that influence greatly the infection outcomes, namely: Age, sex, genetic makeup, social lifestyle factors, population density, and numbers of elderly care facilities.

#### 2.1.1. Age and Sex

Early reports from China showed that COVID-19 affected elderly more with those older than 60 years being the most vulnerable to this infection [7]. Reports from China and Italy suggested high mortality rates, due to COVID-19 in older male patients who had multiple metabolic comorbidities [8,9].

The influence of sex is evident, since more men are affected by the infection. Studies suggest that there are many differences between men and women in the immune response to COVID-19 infection and inflammatory diseases. Women, compared to men, are less susceptible to viral infections based on a different innate immunity, steroid hormones and factors related to sex chromosomes. Testosterone is immunosuppressive in nature, in contrast to the immune-enhancing hormone, estrogen [10]. The presence of two X chromosomes in women enhances the immune system even if one is inactive. The immune regulatory genes encoded by X chromosome in female sex causes lower viral load levels, and less inflammation when compared to male, while a number of CD4 + T cells is higher with improved immune response [11]. In addition, women generally produce higher levels of antibodies which remain in the circulation longer [11]. In Iceland, women and children less than 10 years were found to be less susceptible to COVID-19 infection [12].

#### 2.1.2. Genetic and Racial Characteristics

The genetic makeup of a population is thought to be a crucial factor determining outcome and death/cases ratio. One aspect is related to angiotensin converting enzyme 2 (ACE2), the functional receptor of SARS-CoV-2 in infected cells. Analysis of this receptor suggests that ACE2 is not only a receptor, but is also involved in post-infection regulation, including cytokine production, immune response, and viral genome replication [13].

Another reason for the way the virus attacks people of different racial backgrounds is differences in their genetic makeup. It was found that people with blood group A are more prone to the disease [14]. This probably reflects the availability of the antigen for the entrance of the virus.

Landsteiner’s ABO blood types are carbohydrate epitopes that are present on the surface of human cells. Blood groups are genetically determined; however, environmental factors can potentially influence their distribution in populations. Susceptibility to certain viral infections, such as Norwalk virus and Hepatitis B, has been previously linked to ABO blood group [15,16].

It was also reported that individuals with the O blood group are less susceptible to SARS coronavirus [17]. The ABO blood group distribution in 2173 patients with COVID-19 in Wuhan and Shenzhen, China was compared to that in normal people from the corresponding regions, and it was shown that blood group A was associated with the highest risk for acquiring COVID-19, whereas blood group O was associated with the lowest risk for the infection [18].

It is hypothesized that anti-A antibodies specifically inhibit the adhesion of SARS-CoV S protein-expressing cells to ACE2-expressing cell lines [19]. Consequently, and due to the similarity between SARS-CoV and SARS-CoV-2, the presence of natural anti-blood group antibodies, particularly anti-A antibody, in the blood could explain the link with blood groups.

It is well established now that black people are more prone to the disease essentially because of the high prevalence of existing co-morbidities like hypertension, ischemic heart disease, asthma and diabetes [20]. Their susceptibility to infection is further increased by the intimate personal interaction nature of their jobs and the fact that they live in the southern American states where lockdown was delayed by their governors [20]. Although social conditions, and structural racism may be the main factors in worsening disease outcomes for black people [21], the role of genetics in increasing susceptibility of black people to infection, cannot be over-ruled.

Another factor related to genetic makeup was raised by Cortijo et al. (2020) and Bentrem (2020) [22,23], who drew attention to the fact that the pandemic hit strongly nations in which the R1b haplogroup, characteristic of Western Europe, predominates. This may explain the high fatality rate in some countries like the UK, Spain, and France. Germany is also a country with a similar predominance of R1b haplogroup, however, it exhibited a relatively lower fatality rate (5%), which could be attributed to the advanced healthcare system.

#### 2.1.3. Social Life Style Factors

Human behavioral patterns affect the contact rates between infected and susceptible individuals. Moreover, personal hygiene practices are important in preventing the transmission of infectious diseases. SARS-CoV-2 remains viable in the aerosols for only three hours, however it remains viable on different surfaces for longer durations that may extend for a few days. It is estimated that this virus may remain viable for four hours on copper surfaces, 24 h on cardboard, 48 h on stainless steel surfaces and 72 h on polypropylene plastic [24]. More recently, the airborne potential for transmission of the virus was reported [25].

Sanitary conditions and personal hygiene habits should be considered within the context of the cultural and social standards of any community. A representative example is the populations of the Middle East, North Africa and Africa below Sahara who are largely Muslims. In the Islamic religion practicing Muslims wash specific areas of their body before prayers, or what is called “ablution”, a process which is repeated five times/day. This religious ritual involves washing the face, ears, hands, arms, feet, swirling the mouth, rinsing the nose, and wiping the head with water. The washing of these body organs is repeated three times following teachings of Prophet Muhammad, who also recommended taking a bath at least once weekly, especially before Friday prayers. Taking a bath is mandatory for Muslims after sexual intercourse, and cessation of menstruation at the end of its cycle, also called “Ghosol” which stands for washing the whole body. All of the previous procedures are performed using running water. There are other sanitary practices for Muslims, which mainly originating from religious teachings, include replacing footwear before entering their homes and washing their hands immediately after waking up. There is also an Islamic toilet etiquette which employs the use of water and using left hand for cleaning, as the right hand is used during mealtimes for eating. The latter method of cleaning is important in preventing infections like COVID-19, which are possibly fecal-oral transmitted.

Research that investigates the efficacy of hand hygiene protocols in preventing infection transmission is needed as currently it is mostly focused on health care personnel with minimal regard given to the ordinary population within community settings. The efficacy of using water alone for body cleaning purposes is not well investigated. A study that investigated the efficacy of different handwashing methods to remove feline calicivirus (the respiratory virus that affects cats) from natural and artificial fingernails found that handwashing with water only produced a greater reduction in viral infectious units than a hand sanitizer [26]. It should be noted that hand sanitizers are now recommended in public settings [27], and are used by shoppers upon entering malls and shopping areas.

Behavioral aspects of people play an important role in the spread of disease. People tend to keep at a distance from each other and to observe personal private space of each other in malls, shops and on streets, however, the image is not so perfect when the mass of people residing in the city is high. Even when the population is not so dense in the city, there are instances where social rules for personal private spaces vanish such as in buses, trains, tubes, cinemas, and night clubs. Here, theoretically, there is a higher chance to contract infections.

It was also postulated that social distancing characteristic of Japanese culture (greeting not involving handshaking or kissing), and the frequent use of face masks could contribute to the low infection rate noticed in Japan when compared to other Asian countries [28].

Another aspect that may be considered a cultural characteristic is a diet which should contribute to health and well-being. The link between low immune function and obesity may explain the susceptibility of obese individuals to increased viral pathogenicity [29]. Increased fatty cells may jeopardize the pulmonary microenvironment, including alveoli, consequently contributing to local inflammation and secondary injury associated with viral infection [30]. It is thought that the glycoprotein spikes on SARS-CoV-2 play a key role in host entry and triggering the immune response [31], which is not well established in obese individuals.

On the other hand, the role of vitamin D deficiency in predisposition to severe COVID-19 has been highlighted. Vitamin D is a steroid hormone derived from cholesterol, and it modulates expression of nearly 5% of human genes, some of them are important in boosting the immune response to pathogens [32]. Populations in sunny areas are less likely to have vitamin D deficiencies. The biologically active form of vitamin D (1,25-dihydroxycholecalciferol or calcitriol) has been implicated in the immune response against various inflammatory, infectious, and pulmonary diseases. Indeed, experimental evidence indicates that calcitriol exerts protective effects from lipopolysaccharide-induced lung injury by modulating the expression of ACE 1 and 2 [33]. Augmenting vitamin D status is attributed to physical activity and skeletal muscle contractions, particularly in outdoor settings where the interaction between ultraviolet light and 7-dehydrocholesterol is accomplished in the skin. Directives to limit social contact through lockdown will have an effect of bioavailability of this vitamin. However, even indoor physical activity may effectively improve vitamin D status through biological mechanisms beyond 7-dehydrocholesterol. It was shown that community-dwelling older adults who were followed for over a 2.6 year period, have shown a positive association between serum vitamin D levels and physical activity independent of sun exposure [34]. The link between vitamin D deficiency and COVID-19 severity should be further investigated especially in sunny countries with high rates of cases like Saudi Arabia which is among the top 15 countries with the highest numbers of cases worldwide.

#### 2.1.4. Elderly Home Facilities

Deaths in care homes are much higher than elderly deaths outside. This might be explained on the basis that the spread of disease among this vulnerable section of the population is easier as they are all confined in the same place [35]. The condition is further complicated by the refusal of some workers to attend for work in these facilities, due to their fear of contracting the infection. It was reported that elderly patients in some European countries may have been abandoned in their elderly home facilities, with some being found dead in their beds [36]. Care homes are less popular in the regions where COVID-19 did not hit so hard, as numbers of elderly care homes in many Middle Eastern countries are minimal. Furthermore, elderly people are mostly kept in the care of their families as cultural and religious obligations define the relationships of younger generations with their elders.

#### 2.1.5. Density of Population

One would think that the rate of spread will be proportional to the density of population. This was not the case with SARS-CoV-2, as we can see some of the world’s heavily populated spots (e.g., Gaza strip, Egypt, India, Bangladesh, Indonesia) with lower than expected cases. One would argue that these countries were not first hit by the disease and had time to prepare. Clearly, after a while, they are still underprepared, and people are not displaying high regard to protective etiquettes, such as social distancing, and wearing masks. Within this context, it is interesting to see that two countries with similar population counts have shown different fatality rates. Sweden and Jordan are two countries with an approximately similar population of 10 million; however, Jordan is more densely populated than Sweden with the latter being five times the area of Jordan. However, Sweden had 10 times the fatality rate in Jordan. It was noticed that despite the higher density of population in Jordan, Sweden had adopted more lax procedures than Jordan in combating the spread of COVID-19. This may be one of the factors that contributed to the wide variability in disease outcomes between both countries.

### 2.2. Healthcare Policy

#### 2.2.1. Bacille Calmette-Guérin (BCG) Vaccination

There are countries that still have endemics of tuberculosis, and usually, they would carry out vaccinations against *Mycobacterium tuberculosis* the causative organism of tuberculosis. Some researchers hypothesized that the map of COVID-19 spread is similar to the map of the countries still carrying out Bacille Calmette-Guérin vaccine (BCG) vaccinations inverted. It was observed that the incidence of COVID-19 cases in countries where the BCG vaccine is routinely used in neonates had less reported cases of COVID-19 to date than countries where it is not used. This suggests that this vaccine is also active against COVID-19.

Exploring immunity status against *Mycobacterium tuberculosis* in patients with COVID-19, suggests that COVID-19 patients might have immunity against this microorganism and SARS-CoV-2 as well. The fact that BCG induces trained immunity, has led some researchers to postulate that other vaccines with similar action, like the oral polio vaccine, may also reduce the infectivity of SARS-CoV-2 [37].

While the WHO stated that: “[T]here is no evidence that the (BCG) protects people against infection with COVID-19 virus” [38], some clinical trials are in progress to investigate the possible association between BCG vaccination and protection against COVID-19 [39].

#### 2.2.2. Screening, Testing, and Under-Reporting

Diagnostic methods continued to emerge very quickly after the identification of the disease. Some methods employed PCR techniques, others employed antibody-antigen reaction or tested for antibody titer.

Certain countries used examination kits extensively while others did not have the means or probably did not consider this a priority. The number of tests/million population varied between countries (Table 1).

The importance of testing was emphasized by the WHO from the outset of the pandemic. However, clearly, utilizations differed between countries.

Countries that employed heavy testing protocols, revealed higher numbers of cases, and this probably reflects the high potential for the virus to spread and reflects its high infectivity.

Linear regression analysis was carried out in this study to determine the significance of tests/M as a predictor variable to cases/M. Data were retrieved from (Worldometers, Our world in data, access date 19 June 2020). Data were analyzed using IBM SPSS software for Windows (SPSS version 21 software, Armonk, NY, USA: IBM Corp). Results are presented in Table 2 and Table 3.

For a total of 191 countries that reported the number of tests/M, the average number of carried out tests (628,839) was nearly 14.5 times the average number of confirmed cases (43,798). Moreover, linear regression analysis revealed that the correlation between these two variables was 0.887, and was significant at *p* < 0.000. R^2^ indicated that 78.6% of the variability of numbers of cases could be accounted for by numbers of tests carried out, highlighting the importance of tests to identify cases. However, as it can be seen from unstandardized B weights of the model (0.070) that if the number of tests is increased by 1000, new cases are expected to increase by a total of 70 new cases.

Germany and South Korea are examples of the countries that relied heavily on testing [40]. Surprisingly, Germany showed a fatality ratio (5%) that was much lower than the ratio found in neighboring Italy (around 14%). This may be explained by the heavy testing protocols employed by the German authorities. Testing revealed more of the silent cases that would go otherwise undetected. This might be important as it would greatly modify the fatality/case ratios announced. On the other hand, some heavily populated countries like India and Egypt did not employ large-scale testing. Hence, this could explain the low infection rate reported there.

Under-reporting seems to be another factor influencing total numbers of cases and deaths. It was suggested that some countries would withhold actual numbers of cases and fatalities to maintain order and social peace. On the other hand, other countries are thought to exaggerate their numbers for political reasons. Examples of such theory include the USA which is said to inflate its numbers so as to place more political pressure on other countries, and Iran which may do so to alleviate economic sanctions placed on the country since 1980.

The way the country defines its COVID-19 cases is clearly different from one country to another.

This could be argued as another cause for the variation of cases and fatalities. Probably there are more numbers of cases and fatalities in the population, but the governments do not announce it for different reasons, such as political, and internal stability reasons.

It was striking that China announced that it is modifying counts of the disease on 17th of April, long after it announced no new cases in Wuhan, the initial focus of the disease.

News also emerged around 17th April 2020 that the United Kingdom and Italy did not count fatalities of the disease among care homes although at a certain stage the Italian government asked care homes to take back their residents from hospitals to ease pressure on hospitals.

Some countries do not count COVID-19 cases that were treated or even died at home. In fact, in New York, at-home-death that is usually filed as the unknown cause, quadrupled during the two months from Mid-February to Mid-April 2020.

Almost all affected countries were reluctant, at first, to take action that would affect their respective economies, such as the stopping air travel, placing curfews, implementing hand hygiene, and social distancing rules.

Moreover, as China is the main producer of personal protective equipment (PPE), the scarcity of these equipment after the disease first spread across China, made certain governments reluctant to request people to use masks when leaving their homes. In fact, many governments encouraged preserving PPE resources and limit their use to hospitals. Heads of states and staff appeared on press without masks for the same reason.

Generally, when a country responded promptly against the disease, this reflected as fewer cases and fewer fatalities.

The ability of the health system to detect SARS-CoV-2 infection is a factor that can influence the diagnosis and reporting of cases [5].

However, some countries that were not quick enough to place curfews and social rules to limit the spread of the virus did not complain of high cases or fatalities. India, for instance, initiated the lockdown late March 2020 [41]. Egypt also started its lockdown early March, however, there were no strict measures regarding social distancing and wearing masks. In these countries, there seems to be some uncertainty as to the actual numbers related to the disease, and there is doubt that reporting actually reflects the reality of infections over there.

#### 2.2.3. Insufficient Physical and Human Health Resources

The imbalance between supply and demand for ventilators, due to a shortage in ventilators, has left medical staff forced to decide on who will get ventilators among their needing patients. There were some reports setting guidelines for eligibility of patients to receive treatment, including critically deficient equipment like ventilators [42]. This clearly would increase death rates among the elderly as a vulnerable group of patients and fatalities as a whole. In fact, this practice in itself, may influence death rates among the elderly.

Physicians and nurses in a number of countries like the UK, Egypt and Belgium have threatened to quit, due to lack of adequate PPE, with subsequent risk to spread the infection to them and to their families [36]. A major challenge faced by healthcare systems is spread of infection among healthcare workers, or worse, their death. The number of healthcare workers losing their lives because of COVID-19 is not exactly known especially in some developing countries that lack transparency in reporting. However, it was estimated that by 9th April 2020, there were 9282 health care workers, mostly women, who were infected by COVID-19 [43]. It is essential to implement efficient surveillance for infected healthcare workers, especially in some developing countries where health systems are weakened by insufficient PPE and lack of regulations that protect the rights of affected healthcare workers.

### 2.3. Environmental Factors

The first most obvious factor affecting the spread of the disease is travel. Limiting travel and social distancing comprised basic guidelines of quarantines adopted by most countries to stop the spread of the virus. The emergence and spread of an epidemic occurs not only because of the virulence and transmission pattern of the infective agent, but also because of population movement. Europe, the USA, and China have the busiest and heaviest travel routes worldwide before the spread of the virus. This is true for domestic and international travel within these countries and between them as well.

Wuhan, where the pandemic started, is a city of more than 11 million residents and is connected to other cities in China as well as other cities internationally through direct and indirect flights.

To contain and control infectious diseases, such as COVID-19, in addition to extensive medical treatment, organized efforts need to be carried out to isolate the source of infection, sever transmission routes, and protect vulnerable individuals [44]. This did not happen when the pandemic first appeared in Wuhan. This may explain why Europe and the USA suffered the most from the disease. Actually, the early map of the “hard” spread of the disease with large numbers of cases and fatalities/cases reflect the map of heavy world travel.

Another example is Jordan in which the patient 0 of COVID-19 was a national who arrived back from a business trip to Italy. Another focus of infection formed in the north of the country where a wedding ceremony of a family infected with COVID-19 was held. Later, it was revealed that this family have just arrived from Spain, one of the countries hit hard by the pandemic.

Jordan is considered one of the countries that adopted early measures of lockdown and closing borders against air, land and sea travel. Besides, the country has a meticulous health system adopting testing and investigating sources of virus spread. However, new cases continued to emerge among overseas students who returned home and among truck drivers entering the country through land ports.

Religious pilgrimage can be a factor in disease spread. Among the cities that have religious pilgrimage and that experienced higher rates of the disease are: Makkah, Madinah (in Saudi Arabia) and the Vatican (Italy). Clearly, all these places registered higher numbers of cases. That is why recently, the Saudi government has announced that pilgrimage of 2020 will not be open for pilgrims from abroad, but only for a limited number of residents and Saudi nationals.

Other postulated environmental factors include G networks and climate. Looking at the map of COVID-19 spread revealed similarity with the coverage map of 5G networks. This made some people blame 5G networks and antennae towers for the pandemic by causing cell damage and altering genetic material with the subsequent emergence of the virus from affected cells. Several attacks against 5G installations were reported in some countries. However, there is no evidence to validate this theory.

Climate was also considered as another environmental factor. It was initially thought that the virus is a sensitive RNA virus that is vulnerable to heat, and it will not cause a similar picture of the disease in warm countries as it does in cold ones. First cases of COVID-19 were identified in November–December 2019, pointing to the role of lower temperatures in the transmission of the virus. At the beginning of the pandemic, about 0.6% mortality rates were reported for Latin America. This was attributed to geographic/climatic cause, since these countries are located in the tropical zone [5]. The increased temperature is thought to reduce the transmission of COVID-19 for a certain degree [45].

Relative humidity (RH) affects all infectious droplets with respiratory viruses, independent of their source (respiratory tract or aerosolized from any fluid) and location (in the air or settled on surfaces). RH, therefore, affects all transmission routes but has the most pronounced effect on airborne transmission.

Indoor humidity measurement in New York and the Midwest showed a vapor pressure of below 10 mb or indoor RH of below 24% in winter, a value that favors the stability of winter viruses [46].

It is postulated that the low humidity and high temperature environment would promote the viability of SARS-CoV-2 in the droplets and impaired ciliary clearance and innate immune defense, for robust access to the deep lung tissue and rapid transmission between infected Individuals [46].

The known case-fatality data for COVID-19 in the European countries (mainly Mediterranean), which have four seasons per year, show a high case fatality in Italy, Spain, France, and Greece. On the other hand, the Nordic countries, where the winter season is more extensive (up to 8 months of rain and snow) have shown a fatality rate of less than 1% [5].

It is postulated that a higher frequency of viral respiratory infections, could be related to a persistent exposure and preparation of the innate and adaptive immune responses, which allows an efficient response to respiratory viral infections, with better clinical outcomes, including fewer complications and mortality [5]. It is also postulated that the disease map coincides with the spread of pollution globally, by directly affecting the ability of lungs to remove pathogens, and indirectly by worsening underlying cardiovascular or pulmonary diseases [47]. On the other hand, the correlation between air pollutants like Sulfur oxide (SO_2_) and increased susceptibility to COVID-19 is still controversial and needs more research [48].

However, in Saudi Arabia, late May-early June witnessed very high temperatures beyond 40 °C, with no lowering down of infection rates or fatalities.

### 2.4. Virus Factors

It is evident that the corona family of viruses kept changing through mutations as it first emerged as SARS COV-1 in 2003, then as MERS in Saudi Arabia in 2016 and now as SARS COV-2.

In fact, the viral genome is documented across many affected countries, with a possibility that less vicious strains are affecting countries showing fewer fatalities.

However, the evidence that SARS-CoV-2 has milder strains is considered hypothetical and is still unsubstantiated.

Phylogenetic analysis of SARS-CoV-2 genomes shows that they are highly similar and most contain no more than 10 mutations compared to the virus that started the original outbreak, therefore it is highly unlikely that the virus has evolved a significantly different phenotype [28].

The spread of the virus across the globe in no time suggests that this is the way other viruses, e.g., influenza viruses, might be spreading. However, there are modifying factors, such as the infective dose, which is the number of complete viral bodies capable of causing the disease in a given subject. It appears that infectivity of this virus is very high with a few numbers of the virus capable of initiating the disease in a given human being. It may be helpful to remember that SARS-CoV was responsible for the severe acute respiratory syndrome that occurred in China in November 2002, causing a total of 8096 reported cases, including 774 deaths in 27 countries, yet the SARS pandemic was declared to be over by July 2003 [49]. As for MERS-CoV, there have been 1728 confirmed cases, including 624 deaths in 27 countries [49]. Infectivity of the virus probably might explain why SARS-CoV and MERS died out in place after limited numbers of confirmed cases although their potential for causing fatalities was initially much higher than SARS COV-2.

As a summary, it appears that SARS-CoV-2 is highly infective, and there seems to be several factors that contribute to variability in numbers of cases and deaths/cases.

While genetics, age and medical condition seem to be important factors, the role of healthcare policy is of paramount importance. Evidence suggests that countries that implemented strict measures of social distancing, lockdown, testing and tracking, and maintained these measures throughout the critical period have shown control over the spread of infections. While lockdown appears to have been an effective measure in combating the spread of the disease in countries that implemented it, it is important to highlight the economic consequences, and the difficulty to maintain lockdown. This should be balanced with educating the public to carry out appropriate measures, such as social distancing, going out only when necessary, and working from home.

Moreover, age range in a society seems to influence deaths/cases rates with the following main factors implicated: Proportion of elderly in the society, their medical conditions and the presence of elderly care facilities.

While the role of BCG is still under investigation by many researchers around the world, it should be emphasized that 5G networks have no role in the pandemic.

It is difficult to maintain zero new cases by limiting travel to and from a country. Educating the population on how to continue life activities while cohabiting with the virus is essential.

## Figures and Tables

**Figure 1 healthcare-08-00216-f001:**
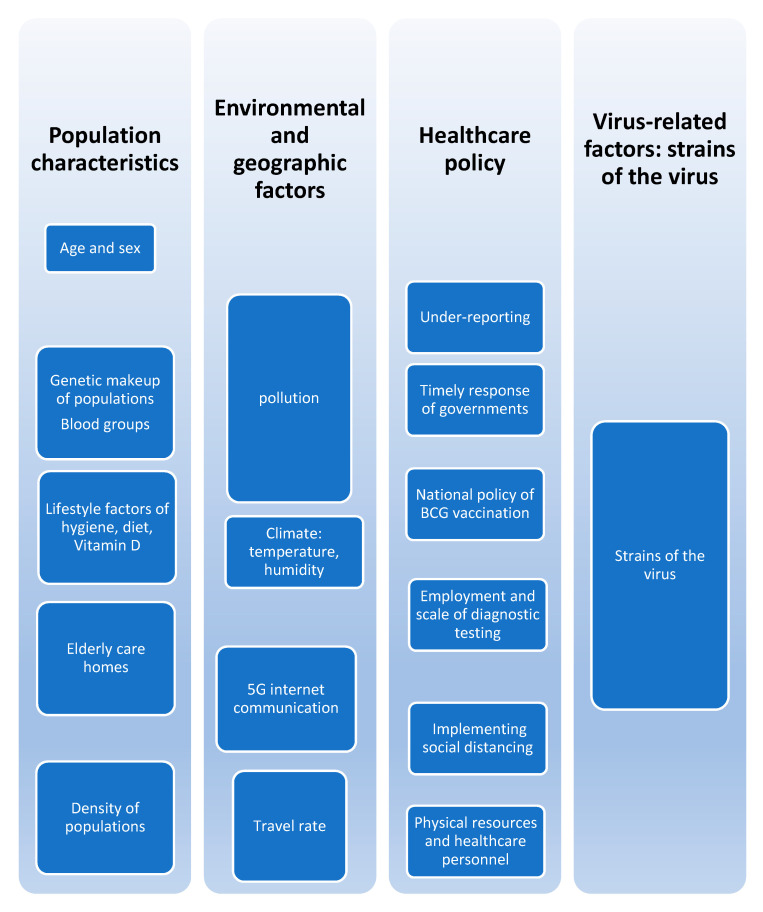
Possible factors implicated in the variability of disease outcomes of infection and fatality among countries.

**Table 1 healthcare-08-00216-t001:** The top 16 affected countries according to the number of cases and deaths.

	Country	Total Cases	Total Deaths	Cases/M	Deaths/M	Total Tests	Tests/M	Population
1	USA	3,159,414	134,867	9544	407	39,482,200	119,265	331,044,624
2	Brazil	1,716,196	68,055	8073	320	4,468,829	21,021	212,591,154
3	India	769,257	21,161	557	15	10,740,832	7782	1,380,270,828
4	Russia	707,301	10,843	4847	74	22,079,294	151,294	145,935,982
5	Peru	312,911	11,133	9488	338	1,842,316	55,862	32,979,917
6	Chile	303,083	6573	15,852	344	1,220,790	63,850	19,119,526
7	Spain	299,593	28,396	6408	607	5,734,599	122,652	46,755,218
8	UK	286,979	44,517	4227	656	11,041,203	162,625	67,893,830
9	Mexico	275,003	32,796	2132	254	684,804	5310	128,958,893
10	Iran	248,379	12,084	2956	144	1,872,391	22,287	84,012,442
11	Italy	242,149	34,914	4005	577	5,754,116	95,173	60,459,584
12	Pakistan	240,848	4983	1090	23	1,491,437	6750	220,955,441
13	South Africa	224,665	3602	3787	61	1,944,399	32,777	59,322,322
14	Saudi Arabia	220,144	2059	6322	59	2,071,823	59,496	34,822,930
15	Turkey	208,938	5282	2477	63	3,782,520	44,840	84,356,463
16	Germany	198,765	9115	2372	109	6,376,054	76,096	83,790,088

Source: Worldometer (https://www.worldometers.info/coronavirus/) accessed 9 July 2020. M, per million.

**Table 2 healthcare-08-00216-t002:** Linear regression analysis of the number of cases/million with tests/million as the predictor variable.

Model	Unstandardized Coefficients		Standardized Coefficients	t	Sig.	95.0% Confidence Interval for B	
	B	Std. Error	Beta			Lower Bound	Upper Bound
(Constant)	955.097	260.348		3.669	0.000	441.536	1468.657
Tests/M	0.023	0.003	0.426	6.470	0.000	0.016	0.030

**Table 3 healthcare-08-00216-t003:** Summary of the regression model.

R	R Square	Adjusted R Square	Std. Error of the Estimate	Durbin-Watson
0.426	0.181	0.177	3057.12969	2.160

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
