# Peer review of "Factors Influencing Global Variations in COVID-19 Cases and Fatalities; A Review"

_healthcare, 2020, doi:10.3390/healthcare8030216_

Round 1

Reviewer 1 Report

This is a narrative review to explore potential factors affects the variations in COVID-19 cases and fatalities across different countries, using a comprehensive literature search (including databases and grey literature). The authors summarised currently available theories and postulations into four main categories, including population characteristics, environmental and geographic factors, healthcare policy, and viral factors.

Major comments

  1. The main issue with this manuscript is the semi-adopted systematic review approach for a review article. There are significant deficiencies in methodology from a systematic review standpoint, although this manuscript can be an informative review article. The authors may consider changing this manuscript into a review article.
    • Methods: It is uncertain whether the identified factors were pre-specified or identified from the literature review. If it is the later, Figure 1 should be moved to the results section, with a summary paragraph.
    • Methods: Given the authors have adopted a methodology similar to a systematic review, details about inclusion criteria (how studies were being selected) and evaluation of study quality should be provided.
  2. Results: A summary paragraph of study characteristics and quality should be provided at the beginning of the Results section.
  3. Many statements have been included, without providing the references (for examples, Line 97-99 regarding the effects of immune regulatory genes encoded by X chromosome and viral load, Line 108-109). It is important to provide all the references.
  4. Given the lack of conclusive supportive evidence for most of these factors, the “Recommendations” section should be removed.

Minor comments

  1. Line 28: Provide the full number for 29K
  2. Line 52: Remove USA and Brazil from the list, given the quoted fatality rate of 6% is within the lower fatality/case rates (4-8%) in the preceding sentence.
  3. Remove “… etc” from the manuscript.
  4. Provide the full term of PPE at its first appearance
  5. Line 334: Typo for “Wohan”
  6. Line 334: Change “This many explains…” to “This may explain…”
  7. Line 349: Typo for “G networks"

Author Response

Reviewer 1

We thank you for your valuable comments. All of them were addressed in the revised manuscript and modifications were highlighted in yellow.

Comments and Suggestions for Authors

This is a narrative review to explore potential factors affects the variations in COVID-19 cases and fatalities across different countries, using a comprehensive literature search (including databases and grey literature). The authors summarised currently available theories and postulations into four main categories, including population characteristics, environmental and geographic factors, healthcare policy, and viral factors.

Major comments

  1. The main issue with this manuscript is the semi-adopted systematic review approach for a review article. There are significant deficiencies in methodology from a systematic review standpoint, although this manuscript can be an informative review article. The authors may consider changing this manuscript into a review article.

Response:Thank you for this valuable comment. We never intended this article to be a systematic review because we had so many variables to tackle which made it difficult to follow the strict methodology of the systematic review.

We already indicated in the original manuscript (abstract line #19-20 ) that this is an overview. To clarify things more, the final paragraph of the introduction which explains the aim of the study (line#60-63) is now modified to explain that this is a review. To remove all confusion we explained the nature of the study in the title; the title now is modified to: “Factors Influencing Global Variations in COVID-19 Cases and Fatalities; A review”

  1. Methods: It is uncertain whether the identified factors were pre-specified or identified from the literature review. If it is the later, Figure 1 should be moved to the results section, with a summary paragraph.

Response: Thank you again. We identified the factors from the literature review, hence figure 1 was moved to results section and a summary paragraph was added according to your suggestion.

  1. Methods: Given the authors have adopted a methodology similar to a systematic review, details about inclusion criteria (how studies were being selected) and evaluation of study quality should be provided.

Results: A summary paragraph of study characteristics and quality should be provided at the beginning of the Results section.

Response: This is a response to the two comments mentioned above because they relate to the same issue.  It is now made clear that this article is a review, and hence these changes may be unrequired.

  1. Many statements have been included, without providing the references (for examples, Line 97-99 regarding the effects of immune regulatory genes encoded by X chromosome and viral load, Line 108-109). It is important to provide all the references.

Response: More references were added to support statements that lacked references. Added references are highlighted in yellow in the reference list.

  1. Given the lack of conclusive supportive evidence for most of these factors, the “Recommendations” section should be removed.

Response: Recommendations section was removed

Minor comments

  1. Line 28: Provide the full number for 29K

Response: The full number for 29K was provided

  1. Line 52: Remove USA and Brazil from the list, given the quoted fatality rate of 6% is within the lower fatality/case rates (4-8%) in the preceding sentence.

Response: USA and Brazil were removed from the list

  1. Remove “… etc” from the manuscript.

Response: “… etc” was removed from text

  1. Provide the full term of PPE at its first appearance

Response: The full term of PPE was provided at its first appearance (highlighted in yellow)

  1. Line 334: Typo for “Wohan”

Response: This was corrected to Wuhan

  1. Line 334: Change “This many explains…” to “This may explain…”

Response: This was corrected

  1. Line 349: Typo for “G networks"

Response: It’s been corrected now

Reviewer 2 Report

Abu Hammad et al performed a review COVID-19 cases, fatalities and proposed possible explanation for that.

While the authors use a somewhat reliable source to define the number of COVID cases and mortality. Nonetheless, due to significant variation in the number of the cases reported, the widespread testing, and the way mortality is counted (death with COVID or death because of COVID), those numbers should not used with extreme caution and limited the validity of this study.

Furthermore, the studies on genetic and racial variations have significant limitations. Also, effect of vitamin D and BCG vaccine on COVID is not really established.

Authors should include a discussion of the quality and biases of the studies they are including.

Author Response

Comments and Suggestions for Authors

Abu Hammad et al performed a review COVID-19 cases, fatalities and proposed possible explanation for that.

While the authors use a somewhat reliable source to define the number of COVID cases and mortality. Nonetheless, due to significant variation in the number of the cases reported, the widespread testing, and the way mortality is counted (death with COVID or death because of COVID), those numbers should not used with extreme caution and limited the validity of this study.

Response: Thank you for your comment. Variations in numbers is an important issue, and it is very noticeable worldwide. Many studies referred to this point. However, as you mentioned: ”it should be used with extreme caution”. That is why we explained this in details in the original manuscript. The issue of “scale of testing” and “underreporting” were mentioned in detail in text to explain that there are many factors that contribute to the variations in cases and deaths.

Furthermore, the studies on genetic and racial variations have significant limitations. Also, effect of vitamin D and BCG vaccine on COVID is not really established.

Response: Thank you for your comment. We totally agree that issues of genetics and racial variations are controversial, and that more evidence is needed. We did not attempt to favor one theory over the other; we just tried to explore available evidence and point out to the strengths and weaknesses so that hopefully more research is conducted towards this direction.

Regarding the role of vitamin D, mounting evidence supports an indirect ( but valid) relationship between vitamin D deficiency and COVID-19 severity. We added recent references to support this link.

Regarding the role of BCG vaccine, it is already stated in text that the role is questionable and ongoing research is being conducted to validate this correlation.

Authors should include a discussion of the quality and biases of the studies they are including.

Response: Thank you.  Requested information is a feature of systematic reviews.  The current manuscript was not intended to be a systematic review.  We referred to various types of studies to retrieve information about the factors associated with disease outcomes including epidemiological studies, clinical trials, case reports, and reviews. We felt that it is probably unfair to perform a systematic review as it was clear that many of the COVID-19 literature were published in an expedited approach. We did our best, though, to refer to reliable sources on the topic.

Reviewer 3 Report

The manuscript is a review on COVID-19. The manuscript does not provide new information, but is is still helpful. The manuscript can be accepted in the present form.

Author Response

The manuscript is a review on COVID-19. The manuscript does not provide new information, but is is still helpful. The manuscript can be accepted in the present form.

Response: Thank you. We hope that this article will be helpful.

Round 2

Reviewer 1 Report

Thank you for the amendment. Please see the following comments:

  1. The current structure of this manuscript is still a mixed of a systematic review and a narrative review. A narrative review does not contain "Methods" and "Results" sections. A thorough re-structuring of the manuscript into a narrative review is needed.  
  2. Paragraph structure needs to be improved. It is unncessary to have multiple paragraphs for 3.2.2, 3.2.3, 3.3.1, 3.3.3, 3.4 and "Conclusions. Some of the paragraphs only consist of a single sentence. 

Author Response

Thank you for the amendment. Please see the following comments:

The current structure of this manuscript is still a mixed of a systematic review and a narrative review. A narrative review does not contain "Methods" and "Results" sections. A thorough re-structuring of the manuscript into a narrative review is needed. 

Response: Thank you for the comment. “Methods” and “Results” headings are now removed, and restructuring was performed so as to go with the structure of a narrative review. Modifications have been highlighted in blue.

Paragraph structure needs to be improved. It is unncessary to have multiple paragraphs for 3.2.2, 3.2.3, 3.3.1, 3.3.3, 3.4 and "Conclusions. Some of the paragraphs only consist of a single sentence.

Response: Thank you. The subsections of testing and under-reporting (formerly 3.2.2, 3.2.3) are now merged in one section. All environmental factors (formerly 3.3.1, 3.3.2, 3.3.3) are now merged in one section. Virus-related factors were discussed as a separate section because it does not fit in other sections. Conclusions heading was removed

Reviewer 2 Report

None

Author Response

Thank you. The manuscript was checked for English language and minor modifications were made